# Proteomic Determination of Low-Molecular-Weight Glutenin Subunit Composition in Aroona Near-Isogenic Lines and Standard Wheat Cultivars

**DOI:** 10.3390/ijms22147709

**Published:** 2021-07-19

**Authors:** Kyoungwon Cho, You-Ran Jang, Sun-Hyung Lim, Susan B. Altenbach, Yong Q. Gu, Annamaria Simon-Buss, Jong-Yeol Lee

**Affiliations:** 1Department of Biotechnology, College of Agriculture and Life Sciences, Chonnam National University, Gwangju 61186, Korea; kw.cho253@gmail.com; 2National Institute of Agricultural Science, RDA, Jeonju 54874, Korea; jang6122@korea.kr; 3Division of Horticultural Biotechnology, Hankyong National University, Anseong 17579, Korea; limsh2@hknu.ac.kr; 4USDA-ARS, Western Regional Research Center, 800 Buchanan Street, Albany, CA 94710, USA; susan.altenbach@usda.gov (S.B.A.); yong.gu@usda.gov (Y.Q.G.); Anna.Simon-Buss@chemie.uni-hamburg.de (A.S.-B.); 5Hamburg School of Food Science, Institute of Food Chemistry, University of Hamburg, 20146 Hamburg, Germany

**Keywords:** LMW-GS, proteomic analysis, RP-HPLC, Aroona near-isogenic lines, standard wheat cultivars

## Abstract

The low-molecular weight glutenin subunit (LMW-GS) composition of wheat (*Triticum aestivum*) flour has important effects on end-use quality. However, assessing the contributions of each LMW-GS to flour quality remains challenging because of the complex LMW-GS composition and allelic variation among wheat cultivars. Therefore, accurate and reliable determination of LMW-GS alleles in germplasm remains an important challenge for wheat breeding. In this study, we used an optimized reversed-phase HPLC method and proteomics approach comprising 2-D gels coupled with liquid chromatography–tandem mass spectrometry (MS/MS) to discriminate individual LMW-GSs corresponding to alleles encoded by the *Glu-A3*, *Glu-B3*, and *Glu-D3* loci in the ‘Aroona’ cultivar and 12 ‘Aroona’ near-isogenic lines (ARILs), which contain unique LMW-GS alleles in the same genetic background. The LMW-GS separation patterns for ‘Aroona’ and ARILs on chromatograms and 2-D gels were consistent with those from a set of 10 standard wheat cultivars for *Glu-3*. Furthermore, 12 previously uncharacterized spots in ‘Aroona’ and ARILs were excised from 2-D gels, digested with chymotrypsin, and subjected to MS/MS. We identified their gene haplotypes and created a 2-D gel map of LMW-GS alleles in the germplasm for breeding and screening for desirable LMW-GS alleles for wheat quality improvement.

## 1. Introduction

Gluten proteins comprise 70−80% of the total wheat flour protein; give wheat dough its unique viscoelastic properties; and make it possible to produce bread, pasta, noodles, and other products [1,2]. Gluten proteins can be separated on the basis of their solubility into monomeric gliadins that dissolve in 60–80% alcohol and polymeric glutenins that are weakly acidic and alkaline-soluble [3,4]. Glutenins consist of high-molecular-weight glutenin subunits (HMW-GSs; 70–90 kDa) and low-molecular-weight glutenin subunits (LMW-GSs; 20–45 kDa) that are polymerized by inter-molecular disulfide bonds [5]. HMW-GSs are encoded by genes at the *Glu-1* loci (*Glu-A1*, *Glu-B1*, and *Glu-D1*) on the long arms of homoeologous group 1 chromosomes, with three to five active genes in most bread wheat cultivars. LMW-GSs are encoded by genes at the *Glu-3* loci (*Glu-A3*, *Glu-B3*, and *Glu-D3*) on the short arms of homoeologous group 1 chromosomes, with the gene copy number varying from 10 to 40 [6,7]. Genetic differences in HMW-GSs and LMW-GSs among wheat cultivars affect the physical properties of dough and determine their end-use qualities [8,9]. Allelic variants of LMW-GSs affect dough strength and extensibility and are far more complicated than those of HMW-GSs that correlate with dough elasticity [10,11,12,13,14].

Glutens can be further divided into the A-, B-, C-, and D-groups according to their mobility in SDS-PAGE based on molecular weight. HMW-GSs and LMW-GSs make up the A- and B-groups, respectively. The C-group contains α/β- and γ-gliadins and some LMW-GSs, while the D-group comprises mostly ω-gliadins and some LMW-GSs [9], indicating that LMW-GSs in the C- and D-groups overlap with gliadins in SDS-PAGE. Most genes encoding LMW-GSs in groups B, C, and D are located close to the *Gli-1* loci encoding γ- and ω-gliadin on the short arms of group 1 chromosomes [15]. Some LMW-GSs in the C-group are located near the *Gli-2* loci encoding α/β-gliadin on the short arms of group 6 chromosomes [16]. The close location of *Gli-1*/*-2* encoding gliadin and *Glu-3* encoding C-/D-group LMW-GSs and their frequent recombination make analyzing the gene structure and function of LMW-GSs difficult [17,18].

Many researchers have studied individual LMW-GS alleles due to their importance in wheat breeding. The allelic composition of LMW-GS has been analyzed by PCR [19], SDS-PAGE [20,21], two-dimensional gel electrophoresis (2-DGE) [22,23], and reversed phase-ultra high-performance liquid chromatography (RP-UHPLC) [20,21,24]. In hexaploid wheat cultivars, six (*a*, *b*, *c*, *d*, *e*, and *f*), nine (*a*, *b*, *c*, *d*, *e*, *f*, *g*, *h*, and *i*), and five (*a*, *b*, *c*, *d*, and *e*) allelic forms at the *Glu-A3*, *Glu-B3*, and *Glu-D3* loci, respectively, have been reported [25]. Four, three, and seven genes were identified at the *Glu-A3d*, *Glu-B3b*, and *Glu-D3c* LMW-GS alleles, respectively, in the Chinese wheat cultivar ‘Xiaoyan 54′ [26]. Using proteomic analysis coupled with 2-DGE and tandem mass spectrometry (MS/MS), one, two, and five LMW-GSs were identified at the *Glu-A3c*, *Glu-B3h*, and *Glu-D3a* in the Korean cultivars ‘Jokyung’ [27] and ‘Keumkang’ [28]. In the Australian and New Zealand cultivars, *Glu-3* alleles have been ranked according to dough resistance [25,29], thereby determining the best combinations of *Glu-3* alleles for dough quality [11] and ranking each allele for dough strength by analyzing the relationship between the allelic variation of *Glu-3* and bread wheat quality [30]. Additionally, Zhang et al. (2012) [19] and Rasheed et al. (2014) [31] used near-isogenic lines from the ‘Aroona’ cultivar to rank each allele at the *Glu-3* loci for dough strength and extensibility, providing information for predicting dough-processing qualities based on the composition of LMW-GS alleles.

Despite efforts to accurately and reliably determine LMW-GS alleles in wheat germplasm, the contributions of LMW-GS alleles to quality have been difficult to assess. This is a result of a lack of efficient techniques for isolating individual LMW-GS genes from the multiple highly similar LMW-GS genes within a cultivar and the allelic variation between cultivars. To identify LMW-GSs corresponding to different alleles encoded by the *Glu-A3*, *Glu-B3*, and *Glu-D3* loci, we fractionated glutenins from ‘Aroona’, a set of 12 ‘Aroona’ near-isogenic lines (ARILs) containing unique LMW-GS alleles in the same genetic background, and a set of 10 standard wheat cultivars. Glutenins were separated using optimized methods for RP-HPLC and 2-DGE combined with LC–MS/MS. We compared the patterns of separated LMW-GSs on chromatograms and 2-D gels and identified peaks and protein spots corresponding to each LMW-GS allele through comparative analysis among ARILs and standard wheat cultivars. The results will be used to identify LMW-GS alleles in germplasm prior to breeding and to screen for desirable LMW-GS alleles for wheat quality improvement.

## 2. Results

### 2.1. Identification of LMW-GS Alleles Using RP-HPLC

To determine the allelic composition of low-molecular-weight glutenin subunits (LMW-GSs) in ‘Aroona’ and 12 ‘Aroona’ near-isogenic lines (ARILs; Table 1), we extracted their glutenin fractions and analyzed them by RP-HPLC. The separation of LMW-GSs was optimized under the following conditions: linear gradient from 25 to 45% solvent B and 0.8 mL/min flow rate for 50 min at 65 °C. We observed five HMW-GS peaks between 5 and 20 min, and LMW-GS and gliadin peaks were detected between 20 and 45 min (Figure 1). HMW-GS alleles in ‘Aroona’ comprise Ax1 at the *Glu-A1* locus, Bx7 and By9 at the *Glu-B1* locus, and Dx2 and Dy12 at the *Glu-D1* locus [32], which can be identified as peaks in RP-HPLC based on hydrophobicity [33]. We identified the peaks corresponding to each LMW-GS encoded at *Glu-3* loci by comparing LC peak patterns in ‘Aroona’ with those in 12 ARILs: six alleles (*a*, *b*, *c*, *d*, *e*, and *f*) at *Glu-A3*, six (*a*, *b*, *c*, *d*, *g*, and *h*) at *Glu-B3*, and three (*b*, *c*, and *f*) at *Glu-D3* (Figure 1).

To identify LMW-GS alleles at the *Glu-A3* locus, we compared LMW-GS peaks in lines ARIL 15-4 (*Glu-A3a*), ARIL 16-1 (*Glu-A3b*), ‘Aroona’ (*Glu-A3c*), ARIL 18-5 (*Glu-A3d*), ARIL 19-2 (*Glu-A3e*), and ARIL 20-1 (*Glu-A3f*). Peaks that were not common to all *Glu-A3* allelic lines are marked with red arrowheads in Figure 1A. We detected two LMW-GS peaks in ARIL 15-4 (*Glu-A3a*), ARIL 16-1 (*Glu-A3b*), and ‘Aroona’ (*Glu-A3c*) that could not be distinguished by RP-HPLC because they had the same retention time. However, we detected five, three, and two unique peaks in ARIL 18-5 (*Glu-A3d*), ARIL 19-2 (*Glu-A3e*), and ARIL 20-1 (*Glu-A3f*), respectively. These results indicate that it is difficult to discriminate the *Glu-A3a*, *-A3b*, and -*A3c* alleles by RP-HPLC but easy to distinguish the *Glu-A3d*, *-A3e*, and -*A3f* alleles.

We analyzed the *Glu-B3* alleles in ‘Aroona’ (*Glu-B3b*) and ARIL 21-2 (*Glu-B3a*), ARIL 23-4 (*Glu-B3c*), ARIL 24-3 (*Glu-B3d*), ARIL 27-6 (*Glu-B3g*), and ARIL 28-4 (*Glu-B3h*). After excluding peaks common to all *Glu-B3* allelic lines, the remaining peaks were marked with blue arrows as shown in Figure 1B. We detected two or four LMW-GS peaks between 25 and 45 min. Two unique LMW-GS peaks were eluted at similar retention times in ‘Aroona’ (*Glu-B3b*) and ARIL 23-4 (*Glu-B3c*). Compared to LMW-GS peak patterns in ‘Aroona’ (*Glu-B3b*), four distinct peaks were detected at different retention times, respectively, in ARIL 21-2 (*Glu-B3a*), ARIL 24-3 (*Glu-B3d*), ARIL 28-4 (*Glu-B3h*), and ARIL 27-6 (*Glu-B3g*). One additional peak (black arrow) eluted between 5 and 25 min was observed in ARIL 27-6 (*Glu-B3g*) and in standard wheat cultivar, ‘Glenlea’ (*Glu-B3g*) (Appendix A).

For LMW-GS alleles at the *Glu-D3* locus, we analyzed ‘Aroona’ (*Glu-D3c*), ARIL 36-2 (*Glu-D3b*), and ARIL 35-1 (*Glu-D3f*), as shown in Figure 1C. Comparing LMW-GS peaks among the three lines, we detected three, four, and four unique peaks (green arrowheads in Figure 1C) in ‘Aroona’ (*Glu-D3c*), ARIL 36-2 (*Glu-D3b*), and ARIL 35-1 (*Glu-D3f*), respectively. LMW-GS peaks in ARIL 36-2 (*Glu-D3b*) and ARIL 35-1 (*Glu-D3f*) showed similar retention times, indicating that it is difficult to discriminate the *Glu-D3b* and -*D3f* alleles using RP-HPLC.

To confirm the accuracy of our LMW-GS identification in ‘Aroona’ and the 12 ARILs, we compared peak patterns of LMW-GSs extracted from 10 standard wheat cultivars (Appendix A). Peaks corresponding to LMW-GS alleles at *Glu-A3*, *Glu-B3*, and *Glu-D3* loci are marked with red, blue, and green arrowheads, respectively, in Appendix A. We detected two unique peaks for *Glu-A3a* and four unique peaks for *Glu-B3a* in ‘Chinese Spring’ (*Glu-A3a*/*Glu-B3a*). LMW-GS peaks in ‘Gabo’ with *Glu-A3b*, *Glu-B3b*, and *Glu-D3b* alleles matched two, two, and four peaks observed in ARIL 16-1 (*Glu-A3b*), ‘Aroona’ (*Glu-B3b*), and ARIL 36-2 (*Glu-D3b*), respectively. Two LMW-GS peaks detected in ‘Aroona’ (*Glu-A3c*) and four detected in ARIL 35-1 (*Glu-D3f*) were also detected in ‘Cheyenne’ (*Glu-A3c* and *Glu-D3f*). Five LMW-GS peaks in ARIL 18-5 (*Glu-A3d*) and four peaks in ARIL 24-3 (*Glu-B3d*) were also observed in the ‘Orca’ cultivar (*Glu-A3d* and *Glu-B3d*). Moreover, three LMW-GS peaks in ARIL 19-2 (*Glu-A3e*), two in ARIL 20-1 (*Glu-A3f*), two in ARIL 23-4 (*Glu-B3c*), four in ARIL 27-6 (*Glu-B3g*), four in ARIL 28-4 (*Glu-B3h*), and three in ‘Aroona’ (*Glu-D3c*) were detected in their standard cultivars ‘Neepawa’ (*Glu-A3e*), ‘Clement’ (*Glu-A3f*), ‘Halberd’ (*Glu-B3c*), ‘Glenlea’ (*Glu-B3g*), ‘Suneca’ (*Glu-B3h*), and ‘Insignia’ (*Glu-D3c*), respectively (Appendix A).

### 2.2. Comparison of LMW-GSs among ‘Aroona’ and Its Near-Isogenic Lines Using 2-DGE

Spot patterns for LMW-GS alleles in 2-DGE were analyzed from ‘Aroona’ and its 12 ARILs. Previously, LMW-GS alleles encoded at the *Glu-3* locus were successfully identified in 32 Korean wheat cultivars and 11 standard wheat cultivars [23]. These results facilitated the identification of LMW-GSs alleles in ‘Aroona’ and its ARILs.

For the identification of *Glu-A3* alleles, LMW-GS fractions were extracted from ‘Aroona’ (*Glu-A3c*), ARIL 15-4 (*Glu-A3a*), ARIL 16-1 (*Glu-A3b*), ARIL 18-5 (*Glu-A3d*), ARIL 19-2 (*Glu-A3e*), and ARIL 20-1 (*Glu-A3f*) and separated on 2-D gels (Figure 2A). Four spots (shown within a black square in Figure 2) at about 43 kDa and 9.2 pI on each gel were used as a reference to compare gel patterns as reported in a previous study [23]. Based on previous 2-DGE results for the LMW-GS encoded by the *Glu-A3c* allele in Korean wheat cultivars ‘Keumkang’ [28] and ‘Jokyung’ [27], we defined the LMW-GS spot (red circle) corresponding to the *Glu-A3c* allele in ‘Aroona’. LMW-GSs from ARIL 15-4 (*Glu-A3a*), ARIL 16-1 (*Glu-A3b*), and ‘Aroona’ (*Glu-A3c*) showed the same patterns in 2-DGE, indicating the difficulty in discriminating the *Glu-A3a*, *-A3b*, and -*A3c* alleles using 2-DGE as well as RP-HPLC. Four and one unique protein spots were detected in 2-DGE of ARIL 18-5 (*Glu-A3d*) and ARIL 20-1 (*Glu-A3f*), respectively, but no distinct spots were observed for ARIL 19-2 (*Glu-A3e*), as shown in Figure 2A.

For the analysis of *Glu-B3* alleles using 2-DGE, we extracted LMW-GSs from ‘Aroona’ (*Glu-B3b*), ARIL 21-2 (*Glu-B3a*), ARIL 23-4 (*Glu-B3c*), ARIL 24-3 (*Glu-B3d*), ARIL 27-6 (*Glu-B3g*), and ARIL 28-4 (*Glu-B3h*). Figure 2B shows two proteins (light blue arrows) within the black square at about 43 kDa and 9.2 pI that were commonly detected in ‘Aroona’ and all the ARILs. Four spots (light blue circles) corresponding to proteins of about 43 kDa with pIs between 9.2 and 10 pI were detected in ‘Aroona’ (*Glu-B3b*) and ARIL 27-6 (*Glu-B3g*). Only two spots in this region were detected in ARIL 21-2 (*Glu-B3a*), while none were detected in ARIL 23-4 (*Glu-B3c*), ARIL 24-3 (*Glu-B3d*), or ARIL 28-4 (*Glu-B3h*). Instead, two, three, and four proteins (light blue circles) with higher molecular weights than those in ‘Aroona’ (*Glu-B3b*) were detected in ARIL 23-4 (*Glu-B3c*), ARIL 24-3 (*Glu-B3d*), and ARIL 28-4 (*Glu-B3h*), respectively (Figure 2B). Moreover, compared to those in ‘Aroona’ (*Glu-B3b*), ARIL 27-6 (*Glu-B3g*) had three additional proteins (spots 9, 10, and 11) on the more basic side.

To identify LMW-GSs at the *Glu-D3* locus, we analyzed ‘Aroona’ (*Glu-3Dc*), ARIL 36-2 (*Glu-D3b*), and ARIL 35-1 (*Glu-D3f*). Two proteins (green arrows within the black square in Figure 2C) derived from *Glu-3Dc* in ‘Aroona’ were detected in all lines. Six LMW-GSs (green circles) in ‘Aroona’ (*Glu-3Dc*) were not detected in ARIL 36-2 (*Glu-D3b*) or ARIL 35-1 (*Glu-D3f*). Three unique proteins were observed in ARIL 36-2 (*Glu-D3b*) and ARIL 35-1 (*Glu-D3f*), indicating that LMW-GSs encoded by *Glu-D3b* and *Glu-D3f* alleles might be highly conserved, making it difficult to distinguish them using 2-DGE.

We previously performed LMW-GS allelic analysis in 11 standard wheat cultivars using 2-DGE [23]. Based on this 2-DGE data and the LMW-GS allelic composition of 10 standard wheat cultivars (Appendix A), we reconfirmed the LMW-GS alleles of ‘Aroona’ and the 12 ARILs (Appendix A). One LMW-GS (red circle) detected in ‘Chinese Spring’ (*Glu-A3a*), ‘Gabo’ (*Glu-A3b*), and ‘Cheyenne’ (*Glu-A3c*) was also observed at a similar gel position among LMW-GSs from ARIL 15-4 (*Glu-A3a*), ARIL 16-1 (*Glu-A3b*), and ‘Aroona’ (*Glu-A3c*), respectively (Figure 2A). Similar to ARIL 18-5 (*Glu-A3d*), ARIL 20-1 (*Glu-A3f*), and ARIL 19-2 (*Glu-A3e*), respectively, four, one, and no LMW-GSs were detected in ‘Orca’ (*Glu-A3d*), ‘Clement’ (*Glu-A3f*), and ‘Neepawa’ (*Glu-A3e*), respectively.

We analyzed LMW-GS alleles at the *Glu-B3* locus in six standard cultivars: ‘Chinese Spring’ (*Glu-B3a*), ‘Gabo’ (*Glu-B3b*), ‘Halberd’ (*Glu-B3c*), ‘Orca’ (*Glu-B3d*), ‘Suneca’ (*Glu-B3b*), and ‘Glenlea’ (*Glu-B3g*). Unique LMW-GS spots in 2-DGE of ARIL 21-2 (*Glu-B3a*), ‘Aroona’ (*Glu-B3b*), ARIL 23-4 (*Glu-B3c*), ARIL 24-3 (*Glu-B3d*), ARIL 27-6 (*Glu-B3g*), and ARIL 28-4 (*Glu-B3h*) were similarly positioned on 2-DGE gels of the corresponding standard cultivars, respectively (Appendix A). Moreover, for *Glu-D3* alleles, three LMW-GSs in ARIL 36-2 (*Glu-D3b*) and six in ‘Aroona’ (*Glu-D3c*) were also detected in ‘Gabo’ (*Glu-D3b*) and ‘Insignia’ (*Glu-D3c*), respectively; however, one of three spots detected in 2-DGE of ARIL 35-1 (*Glu-D3f*) was not observed in the 2-DGE of ‘Cheyenne’ (*Glu-D3f*).

### 2.3. Identification of LMW-GSs Using LC–MS/MS Analysis

We found that the annotation of 24 LMW-GS spots in the 2-DGE of ‘Aroona’, ARILs, and standard wheat cultivars did not match that from our previous study on LMW-GS alleles in standard wheat cultivars and Korean wheat cultivars [23]. To identify mismatched spots, they were excised from stained 2-DGE gels, digested in-gel with either chymotrypsin, thermolysin, or trypsin (‘Aroona’ and ARILS) or only chymotrypsin (standard and Korean cultivars) and then subjected to tandem mass spectrometry (LC–MS/MS) (Table 2 and Appendix A). For LMW-GS alleles encoded at the *Glu-3B* locus, we identified two proteins (spots 1 and 2) in ‘Aroona’ (*Glu-B3b*), two (spots 22 and 23) in ‘Gabo’ (*Glu-B3b*), five (spots 7, 8, 9, 10, and 11) in ARIL 27-6 (*Glu-B3g*), and five (spots 24, 25, 26, 27, and 28) in ‘Glenlea’ (*Glu-B3g*). Spot 1 from the 2-DGE of ‘Aroona’ (*Glu-B3b*) was an LMW-GS encoded by *GluB3-23*, while spot 2 for ‘Aroona’ (*Glu-B3b*) and spots 22/23 for ‘Gabo’ (*Glu-B3b*) were LMW-GSs encoded by *GluB3-22*. In the case of LMW-GS alleles encoded by *Glu-B3g*, we identified spot 7/8 for ARIL 27-6 (*Glu-B3g*) and spots 24/25 for ‘Glenlea’ (*Glu-B3g*) as LMW-GSs encoded by *GluB3-23*, and spot 10 for ARIL 27-6 (*Glu-B3g*) and spots 26/27/28 for ‘Glenlea’ (*Glu-B3g*) as LMW-GSs encoded by *GluB3-15*. Moreover, spots 9 and 11 in 2-DGE of ARIL 27-6 (*Glu-B3g*) were identified as LMW-GSs encoded by *GluB3-16*, and -*51*, respectively.

We used one LMW-GS (spot 12) in ARIL 36-2 (*Glu-D3b*), one (spot 21) in ‘Gabo’ (*Glu-D3b*), four (spots 3, 4, 5, and 6) in ‘Aroona’ (*Glu-D3c*), and four (spots 29, 30, 31, and 32) in ‘Insignia’ (*Glu-D3c*) to identify LMW-GS alleles at the *Glu-3D* locus (Figure 2 and Appendix A). Four proteins (spots 3, 4, 5, and 6) in ‘Aroona’ (*Glu-D3c*) and four (spots 29, 30, 31, and 32) in ‘Insignia’ (*Glu-D3c*) were identified as LMW-GSs encoded by *GluD3-32*. Spot 12 from ARIL 36-2 (*Glu-D3b*) and spot 21 from ‘Gabo’ (*Glu-D3b*) were identified as LMW-GSs encoded by *GluD3-31*, indicating that the *Glu-D3b* and *Glu-D3c* alleles represent LMW-GSs encoded by *GluD3-31* and *GluD3-32*, respectively (Table 2 and Appendix A).

## 3. Discussion

LMW-GSs are encoded by genes at *Glu-3* loci (*Glu-A3*, *Glu-B3,* and *Glu-D3*), with copy numbers varying from 10–40. LMW-GS alleles at the *Glu-3* loci determine end-use quality. To evaluate the dough-processing qualities of wheat cultivars, LMW-GS alleles at the *Glu-3* loci have been ranked for dough strength in wheat cultivars from Australia [25] and New Zealand [29] and for the ‘Aroona’ near-isogenic lines (ARILs) [19,31]. Since the composition of LMW-GS alleles in individual cultivars has been given priority during wheat breeding, SDS-PAGE, 2-DGE, HPLC, and LMW-GS molecular marker systems have been used to define LMW-GS alleles in different wheat cultivars. Previously, one to two peaks corresponding to LMW-GSs encoded at each Glu-3 locus were identified by the composition analysis of LMW-GS in common wheat cultivars using a RP-UHPLC method [24]. In this study, two to five LMW-GS peaks in RP-HPLC corresponding to *Glu-A3*, *-B3*, and -*D3* alleles were determined through comparative analysis of ‘Aroona’ and the 12 ARILs. The accuracy was confirmed by comparing the resulting RP-HPLC patterns with patterns in standard wheat cultivars and suggests that our improved RP-HPLC method can be used to identify LMW-GS alleles faster, easier, and more accurately than traditional methods such as SDS-PAGE. However, three *Glu-A3* alleles (*Glu-A3a*, *-A3b*, and *-A3c*) and two *Glu-D3* alleles (*Glu-D3b* and -*D3f*) could not be discriminated. Furthermore, by integrating our 2-DGE data on LMW-GSs in this study with the composition of LMW-GS genes previously characterized in ‘Aroona’ and ARILs [19,28,36], we created reference maps (Figure 3) for protein spots corresponding to each LMW-GS allele in three standard cultivars, ‘Gabo’ (*Glu-A3b*, *-B3b*, and -*D3b*), ‘Aroona’ (*Glu-A3c*, *-B3b*, and -*D3c*), and ‘Orca’ (*Glu-A3d*, *-B3d*, and -*D3c*).

### 3.1. Glu-A3 Alleles

We identified 4–6 genes at the *Glu-A3* locus in ‘Aroona’ (*Glu-A3c*) and *Glu-A3* ARILs using an LMW-GS gene molecular marker system [19,31,37]. Except for ARIL18-5 (*Glu-A3d*) with three active LMW-GS genes (*GluA3-4*, *GluA3-23*, and *A3-568*), the other lines have one active gene: *GluA3-11* in ARIL15-4 (*Glu-A3a*), *GluA3-12* in ARIL16-1 (*Glu-A3b*), *GluA3-13* in ‘Aroona’ (*Glu-A3c*), *GluA3-15*/*17* in ARIL19-2 (*Glu-A3e*), and *GluA3-16* in ARIL20-1 (*Glu-A3f*) as shown in Table 3. We also previously analyzed the composition of LMW-GS alleles in 32 Korean wheat cultivars and 11 standard wheat cultivars using 2-DGE and SDS-PAGE [23,27]. One LMW-i type subunit encoded by the *GluA3-13* (*A3-620*) gene was identified in ‘Jokyung’ (*Glu-A3c*) and ‘Cheyenne’ (*Glu-A3c*). By comparing LMW-GS spot patterns with those from ‘Jokyung’ [27], we defined one LMW-i type subunit (red circle in Figure 1A) corresponding to GluA3-13 in ‘Aroona’ (*Glu-A3c*).

The retention times of LMW-GSs corresponding to *Glu-A3* alleles in the RP-HPLC analysis (Figure 1A) and their spot patterns in 2-DGE (Figure 2A) were similar among ARIL 15-4 (*Glu-A3a*), ARIL 16-1 (*Glu-A3b*), and ‘Aroona’ (*Glu-A3c*). Similar results were observed among the standard cultivars ‘Chinese Spring’ (*Glu-A3a*), ‘Gabo’ (*Glu-A3b*), and ‘Cheyenne’ (*Glu-A3c*) (Appendix A), indicating that it is difficult to discriminate LMW-GSs derived from *Glu-A3a*, *Glu-A3b*, and *Glu-A3c* alleles using either RP-HPLC or 2-DGE methods. In ‘Aroona’ near-isogenic lines, the *Glu-A3a* allele in ARIL 15-4 was donated by ‘Chinese Spring’, where one gene (*GluA3-11*) at *Glu-A3a* encoding an LMW-i type subunit is actively expressed [38]. The *Glu-A3b* allele in ARIL 16-1 comes from ‘Gabo’, where one gene (*GluA3-12*) at *Glu-A3b* encoding an LMW-i type subunit is expressed [19]. The LMW-GSs encoded by *Glu-A3* alleles in ARIL 15-4 (*Glu-A3a*), ARIL 16-1 (*Glu-A3b*), and ‘Aroona’ (*Glu-A3c*) consist of 356, 364, and 356 amino acid residues excluding signal peptides, respectively. They contain ISQQQQ N-terminal sequences and have 99.44–99.72% identity to each other, resulting in difficulty in distinguishing them.

Through the comparative analysis of RP-HPLC patterns for LMW-GS fractions from ARIL 18-5 (*Glu-A3d*), ARIL 19-2 (*Glu-A3e*), and ARIL 20-1 (*Glu-A3f*), we identified five, three, and two unique peaks, respectively (Figure 1A). Four and one unique protein spots were observed in 2-DGE for ARIL18-5 (*Glu-A3d*) and ARIL20-1 (*Glu-A3f*), respectively, but no distinctly different spot was observed for ARIL19-2 (*Glu-A3e*). We observed similar results for the standard cultivars, ‘Orca’ (*Glu-A3d*), ‘Neepawa’ (*Glu-A3e*), and ‘Clement (*Glu-A3f*), as shown in Appendix A. Using the LMW-GS gene molecular marker system, Zhang et al. (2012) [19] determined that the *Glu-A3d* allele represents three actively expressed genes, *GluA3-4* (*A3-662*), *GluA3-23* (*A3-402*), and (*A3-568*) and that the *Glu-A3f* allele corresponds to one expressed gene, *GluA3-16* (*A3-573*), encoding an i-type LMW-GS with 336 amino acid residues, 39.04 kDa, and ISQQQQQP N-terminal. In this study, we observed one unique LMW-GS spot at about 39 kDa in 2-DGE of ARIL 20-1 (*Glu-A3f*), as shown in Figure 2A, indicating that the spot could be derived from *GluA3-16* (*A3-573*). Moreover, ARIL 18-5 (*Glu-A3d*) possessed three actively expressed LMW-GS genes, encoding GluA3-23 (A3-402)/A3-568/ GluA3-4 (A3-662), in which the signal peptide-truncated forms have 284/338/370 amino acid residues, 32.22/39.28/43.02 kDa, and MDTSCIP/ISQQQQPP/ISQQQQQP N-terminal sequences, respectively. We observed four unique spots, two at 43 kDa and two at 39 kDa, in the 2-DGE patterns of ARIL 18-5 (*Glu-A3d*) compared to those of ‘Aroona’ (*Glu-A3c*), as shown in Figure 2A, indicating that these spots could be GluA3-4 (A3-662) and A3-568, respectively, based on predicted molecular weight and comparative analysis with ‘Aroona’ and the ARILs.

### 3.2. Glu-B3 Alleles

Active genes encoding m- and s-type LMW-GS genes at the *Glu-B3* locus of ‘Aroona’ (*Glu-B3b*) and five ARILs (*Glu-B3a*, -*B3c*, -*B3d*, -*B3g*, and -*B3h*) were identified by using the LMW-GS gene molecular marker system [19,34]. The *Glu-B3* alleles were divided into two groups according to the composition of s-type genes, one group comprising *Glu-B3a*, -*B3b*, and -*B3g* and the other comprising *Glu-B3c*, -*B3d*, and -*B3h*. The *Glu-B3a*, -*B3b*, and -*B3g* alleles correspond to three actively expressed LMW-GS genes, *GluB3-11*/*-21*/*-44*, *GluB3-12*/*-22*/*-43*, and *GluB3-15*/*-23*/*-41*, respectively, as shown in Table 3, while the *Glu-B3c*, -*B3d*, and -*B3h* alleles represent two actively expressed genes, *GluB3-34*/*-44*, *GluB3-31–33*/*-44*, and *GluB3-61–64*/*-43*, respectively. Proteins encoded by LMW-GS genes in each group showed similar electrophoretic patterns in 2-DGE [19]. In this study, we also observed that LMW-GSs from ARIL 21-2 (*Glu-B3a*), ‘Aroona’ (*Glu-B3b*), and ARIL 27-6 (*Glu-B3g*) showed similar patterns in 2-DGE while those from ARIL 23-4 (*Glu-B3c*), ARIL 24-3 (*Glu-B3d*), and ARIL 28-4 (*Glu-B3h*) were similar (Figure 2B).

ARIL 28-4 (*Glu-B3h*) has two active genes: *GluB3-43* (*B3-530b*) and *GluB3-61–64* (*B3-688c*) [19]. Previously, we reported that six protein spots were derived from the *Glu-B3h* allele in Korean wheat cultivar ‘Jokyung’ [27]. Of these six, two encoded by *GluB3-43* were m-type LMW-GSs with a METSHIP N-terminal, corresponding to B3-530b, and four encoded by *GluB3-33* were s-type LMW-GSs with a MENSHIP N-terminal, identical to B3-688b and sharing 98.92% identity with B3-688c. Thus, compared with ‘Jokyung’ (*Glu-B3h*), two spots (light blue arrows in the positioning box in Figure 2B) from ARIL28-4 (*Glu-B3h*) represent m-type LMW-GSs encoded by *GluB3-43* (*B3-530b*). The other four spots (light blue circles) could be identical to *s*-type LMW-GSs encoded by *GluB3-61–64* (*B3-688c*). Furthermore, two spots corresponding to GluB3-43 (B3-530b), light blue arrows in the positioning box in Figure 2B, in ARIL28-4 (*Glu-B3h*) were observed in ARIL 21-2 (*Glu-B3a*), ‘Aroona’ (*Glu-B3b*), ARIL 23-4 (*Glu-B3c*), ARIL 24-3 (*Glu-B3d*), and ARIL 27-6 (*Glu-B3g*). These lines possess one of the *GluB3-41*/*-43*/*-44* genes encoding an m-type LMW-GS consisting of 330 amino acids, which have 99.09–99.39% identity to each other. These results indicate that two spots (light blue arrows) in a square box in Figure 2B are derived from a gene encoding B3-530.

ARIL 27-6 (*Glu-B3g*) shared four spots (light blue circles) with ‘Aroona’ (*Glu-B3b*) and had three unique spots (Figure 2B). The spot patterns were similar to those of their standard cultivars, ‘Gabo’ (*Glu-B3b*) and ‘Glenlea’ (*Glu-B3g*), respectively. Three unique proteins (spots 9, 10, and 11) in ARIL 27-6 (*Glu-B3g*) and three (spots 26, 27, and 28) in ‘Glenlea’ (*Glu-B3g*) were identified as s-type LMW-GSs (ABY58126 or ACZ59817) encoded by the *GluB3-15* (*B3-544*) gene (Table 2 and Appendix A). In ARIL 27-6 (*Glu-B3g*), three LMW-GS genes have been reported to be actively expressed, *GluB3-15* (*B3-544*), *GluB3-23* (*B3-621*), and *GluB3-41* (*B3-530c*) [19], indicating that the four common spots at 42 kDa in the 2-DGE of ARIL 27-6 (*Glu-B3g*) and ‘Aroona’ (*Glu-B3b*) could be derived from a gene encoding B3-621. Indeed, we identified two proteins (spots 1 and 2) in the 2-DGE of ‘Aroona’ (*Glu-B3b*) and two (spots 22, and 23) in ‘Gabo’ (*Glu-B3b*) as s-type LMW-GSs encoded by the *GluB3-22* (*B3-621a*) gene haplotype (Table 2 and Appendix A). The two LMW-GSs (blue circles in Figure 2B) in ARIL 21-2 (*Glu-B3a*), four in ‘Aroona’ (*Glu-B3b*), and four in ARIL 27-6 (*Glu-B3g*) could be GluB3-21 (B3-624), GluB3-21 (B3-621a), and GluB3-23 (B3-621b), respectively; however, the spots in the 2-DGE of ARIL 21-2 (*Glu-B3a*) and ‘Aroona’ (*Glu-B3b*) could also be derived from the *GluB3-11* (*B3-593*) and *GluB3-12* (*B3-607*) genes, respectively.

ARIL 23-4 (*Glu-B3c*), ARIL 24-3 (*Glu-B3d*), and ARIL 28-4 (*Glu-B3h*) have two active genes, *GluB-34*/*-44*, *GluB-31–33*/*-44*, and *GluB-61–64*/*-43*, respectively. These lines and their standard cultivars showed similar protein spot patterns in 2-DGE (Figure 2B, and Appendix A). Compared to ‘Aroona’ (*Glu-B3b*), ARIL 23-4 (*Glu-B3c*), ARIL 24-3 (*Glu-B3d*), and ARIL 28-4 (*Glu-B3h*) had two common spots (light blue arrows within black square in Figure 2B) encoded by one of the *GluB-41*/*-44* genes. ARIL 23-4 (*Glu-B3c*), ARIL 24-3 (*Glu-B3d*), and ARIL 28-4 (*Glu-B3h*) contain two, three, and four unique proteins, respectively, indicating that these could be encoded by *GluB-34*, *GluB-31–33*, and *GluB-61–64*, respectively.

### 3.3. Glu-D3 Alleles

At the *Glu-D3* locus, six active genes were identified in ARIL 36-2 (*Glu-D3b*), Aroona (*Glu-D3c*), and ARIL 35-1 (*Glu-D3f*), including *GluD3-6* (*D3-385*), *GluD3-4* (*D3-394*), *GluD3-22*/*-23* (*D3-441*/*-432*), *GluD3-12* (*D3-528*), *GluD3-5* (*D3-575*), and *GluD3-31*/*-32* (*D3-578b*/*a*) [19,35]. We previously identified eight LMW-GS protein spots encoded by five gene haplotypes using 2-DGE in ‘Jokyung’ (*Glu-D3a*) and ‘Keumkang’ (*Glu-D3a*): *GluD3-6* (*D3-385*), *GluD3-21* (*D3-441*), *GluD3-11* (*D3-525*), *GluD3-5* (*D3-575*), and *GluD3-31* (*D3-578b*); LMW-GS encoded by *GluD3-4* (*D3-394*) was not identified due to an insufficient amount of protein [27,28]. Moreover, ‘Aroona’ (*Glu-D3c*) is known to contain six active LMW-GS genes: *GluD3-6* (*D3-385*), *GluD3-4* (*D3-394*), *GluD3-23* (*D3-432*), *GluD3-12* (*D3-528*), *GluD3-5* (*D3-575*), and *GluD3-32* (*D3-578a*). In this study, we found 10 protein spots encoded by LMW-GS genes at the *Glu-D3c* locus in ‘Aroona’. Four of these were shown to have similar positions on 2-DGE gels compared to those identified in ‘Jokyung’ (*Glu-D3a*) and ‘Keumkang’ (*Glu-D3a*): GluD3-6 (D3-385), GluD3-11 (D3-525), and GluD3-5 (D3-575). We identified four protein spots (spot 3, 4, 5, and 6) in ‘Aroona’ (*Glu-D3c*) and four (spot 29, 30, 31, and 32) in ‘Insignia’ (*Glu-D3c*) encoded by the *GluD3-32* (*D3-578a*) haplotype. Although the remaining two spots were not identified by MS/MS analysis, considering that GluD3-4 (D3-394) is insufficient amount for our 2-DGE system [27,28], we predicted them to be GluD3-23 (D3-432) based on the genetic marker system and comparative analysis.

ARIL 36-2 (*Glu-D3b*) and ARIL 35-1 (*Glu-D3f*) have six active LMW-GS genes such as *GluD3-6* (*D3-385*), *GluD3-4* (*D3-394*), *GluD3-22* (*D3-441*), *GluD3-12* (*D3-528*), *GluD3-5* (*D3-575*), and *GluD3-31* (*D3-578b*). These ARILs and their corresponding standard wheat cultivars, ‘Gabo’ (*Glu-D3b*) and ‘Cheyenne’ (*Glu-D3f*), showed similar protein spot patterns in 2-DGE. Compared with those in ‘Aroona’ (*Glu-D3c*) and ‘Insignia’ (*Glu-D3c*), we observed two unique protein spots at 35 kDa and one at 43 kDa between 9.2 and 10.5 pI in 2-DGE of ARIL36-2 (*Glu-D3b*) and ARIL35-1 (*Glu-D3f*) in Figure 2C. The two spots at 35 kDa were previously observed in 2-DGE of ‘Jokyung’ (*Glu-D3a*) and are encoded by the *GluD3-21* (*D3-441*) haplotype. The other spot at 43 kDa, spot 12 for ARIL 36-2 (*Glu-D3b*) and spot 21 for ‘Gabo’ (*Glu-D3b*), was identified as a LMW-GS encoded by the *GluD3-31* (*D3-578b*) haplotype.

Finally, based on an integrative analysis between the activated LMW-GS genes and 2-DGE separation of LMW-GSs in ‘Aroona’ and its ARILs, we tried to designate the activated genes corresponding to LMW-GS spots separated on 2-DGE in ‘Aroona’ (*Glu-A3c*/-*B3b*/-*C3c*) and two standard cultivars, ‘Gabo’ (*Glu-A3b*/-*B3b*/-*C3b*) and ‘Orca’ (*Glu-A3d*/-*B3d*/-*C3c*), as shown in Figure 3. The results might be used to identify LMW-GS alleles in germplasm prior to breeding and to screen for desirable LMW-GS alleles in wheat quality improvement.

## 4. Materials and Methods

### 4.1. Plant Materials

‘Aroona’ and its 12 near-isogenic lines differing at the *Glu-A3*, *Glu-B3*, and *Glu-D3* loci (Table 1) were provided by Dr. Marie Appelbee and Prof. Ken Shepherd, SARDI Grain Quality Research Laboratory, South Australia. The grains of 10 standard wheat cultivars used for the identification of *Glu-**3* alleles (Appendix A) were kindly provided by the National Bioresource Project-Wheat, Japan (NBRP-Wheat, https://shigen.nig.ac.jp/wheat/komugi/, 12 October 2015) and by the National Plant Germplasm System of the USDA-ARS, USA (NPGS, https://www.ars-grin.gov/npgs/, 12 October 2015). The grains of the wheat cultivars were ground with a mortar and pestle to fine powders and stored at −80 °C until glutenin extraction.

### 4.2. Glutenin Extraction

Glutenin was extracted as previously reported [23]. Wheat flour (300 mg) was mixed with 15 mL of 50% propanol (*v*/*v*) at 65 °C for 30 min and then centrifuged at 10,000× *g* for 5 min. The supernatant fraction containing gliadin was removed. This extraction was repeated three times to minimize gliadin contamination. The precipitate was suspended in 1.5 mL of 50% (*v*/*v*) propanol, 80 mM Tris-HCl (pH 8.0), and 1% (*w*/*v*) dithiothreitol (DTT) at 65 °C for 30 min. After centrifugation at 10,000× *g* for 5 min, 1.5 mL of 80 mM Tris-HCl (pH 8.0) with 1.4% 4-vinylpyridine (*v*/*v*) was added for alkylation and incubated at 65 °C for 15 min. After centrifugation at 10,000× *g* for 2 min, the supernatant was transferred to a new 1.5 mL tube and stored at −4 °C overnight. Glutenin fractions were precipitated using acetone containing 15% TCA (*v*/*v*) and stored at −20 °C until use.

### 4.3. Two-Dimensional Gel Electrophoresis (2-DGE)

For 2-DGE, precipitated glutenin samples were washed with chilled acetone, centrifuged at 12,000× *g* for 10 min, and then dried. The glutenin pellets were dissolved completely using 70 μL of dehydration buffer (7 M urea, 2 M thiourea, 2% (*w*/*v*) CHAPS, and 0.5% (*v*/*v*) IPG buffer (GE Healthcare Life Sciences, USA) and 12.8 μL of 1 M DTT. Total protein was quantified using the Bradford method [39]. Fifty micrograms of glutenin was resolved in 350 μL rehydration solution containing 12.8 μL of 1 M DTT, applied on an 18 cm IPG strip (pH 6–11, GE Healthcare Life Sciences, USA) and then rehydrated in-gel for a total run time of 80 kVh at 20 °C using the IPGphore system (Amersham Biosciences, GE Healthcare Life Sciences, USA). The IPG strips were equilibrated using 6 M urea, 75 mM Tris-HCl (pH 8.8), 29.3% (*v*/*v*) glycerol, 2% (*w*/*v*) SDS, and 1% (*w*/*v*) DTT for 15 min and then incubated with 6 M urea, 75 mM Tris-HCl (pH8.8), 29.3% (*v*/*v*) glycerol, 2% (*w*/*v*) SDS, and 2.5% (*w*/*v*) iodoacetamide for 15 min. The 2-DGE (12.5% acrylamide gel) was carried out at 70 V for 17 h and at 130 V for 6 h. The gel was stained with Coomassie Brilliant Blue R-250 for 3 h, de-stained with a water–methanol–acetic acid buffer (80/10/10, by volume) twice for 3 h each, and then scanned using an Epson Perfection V800 Photo scanner (Epson, Japan). Finally, the patterns of protein spots separated on the gels were analyzed using Image Master Platinum v6.0 (GE Healthcare Life Sciences, Piscataway, NJ, USA).

### 4.4. Identification of LMW-GSs Using UPLC–MS/MS

Protein spots from ‘Aroona’ and 12 ARILs were excised from 2-D gels; reduced; alkylated; digested with trypsin, chymotrypsin, and thermolysin; and then analyzed as previously described [40]. The protein spots from 10 standard wheat cultivars were excised from 2-D gels and digested in-gel with chymotrypsin. The digested peptides were subjected to nanoAcquity UPLC coupled with MS (Waters Synapt G1 HDMS, MA, USA) to obtain their mass spectra. The data were analyzed using a database containing 63,245 *Triticum* protein sequences from NCBI (www.ncbi.nlm.nih.gov, 12 October 2015) and 151,173 *Triticum* protein sequences from UniProt (www.uniprot.org, 12 October 2015). UPLC–MS/MS analysis was conducted at the NICEM (National Instrumentation Center for Environmental Management, Seoul National University, Korea).

### 4.5. Separation of LMW-GSs Using RP-HPLC

An analysis of the LMW-GSs using RP-HPLC was performed using a Waters Alliance e2695 (USA) equipped with an XBridge Protein BEH-C4 column (3.5 m, 4.6 × 250 nm i.d., Waters, USA) following the method of Yu et al. (2013) with modifications [24]. The precipitated glutenin samples were completely dissolved in 500 μL of 0.1% (*v*/*v*) trifluoroacetic acid with 20% (*v*/*v*) acetonitrile and filtered using a PVDF syringe filter (0.45 m, Whatman, Maidstone, UK). Ten microliters of each sample were subjected to RP-HPLC with the following solvents: water with 0.1% trifluoroacetic acid (solvent A) and 20% (*v*/*v*) acetonitrile with 0.1% trifluoroacetic acid (solvent B). The proteins were eluted using a linear gradient from 25 to 45% of solvent B for 50 min under a 0.8 mL/min flow rate and 65 °C column oven temperature and monitored at 206 nm wavelength.

## Figures and Tables

**Figure 1 ijms-22-07709-f001:**
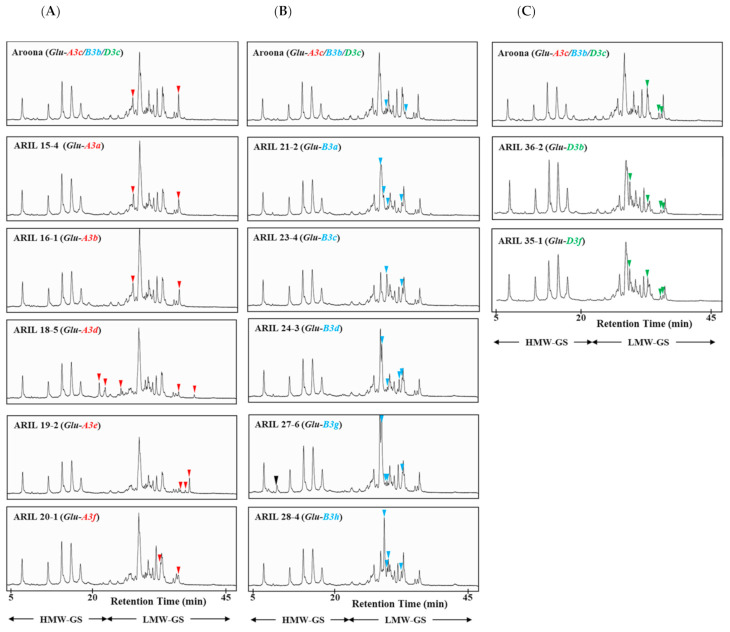
RP-HPLC analysis of LMW-GS fractions in ‘Aroona’ and ARILs. Peaks corresponding to each LMW-GS allele encoded by *Glu-A3* (**A**), *Glu-B3* (**B**), and *Glu-C3* (**C**) are indicated by red, light blue, and green arrowheads, respectively.

**Figure 2 ijms-22-07709-f002:**
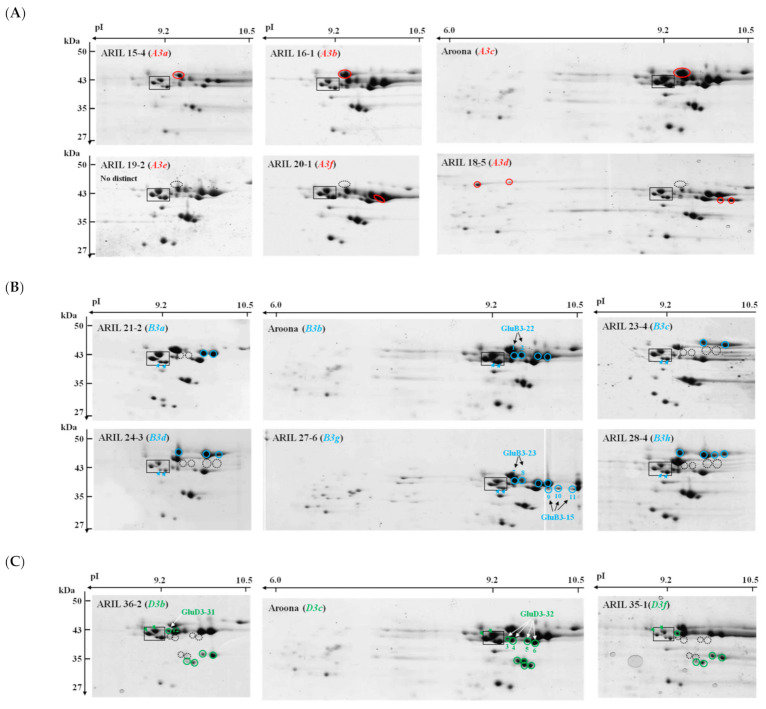
Two-dimensional gel electrophoresis (2-DGE) analysis of LMW-GSs from ‘Aroona’ and its ARILs. Protein spots corresponding to each LMW-GS allele encoded by *Glu-A3* (**A**), *Glu-B3* (**B**), and *Glu-C3* (**C**) are indicated by red, light blue, and green arrows, respectively. Dashed black circles indicate the absence of spots detected in ‘Aroona’. Numbered spots were identified using LC–MS/MS.

**Figure 3 ijms-22-07709-f003:**
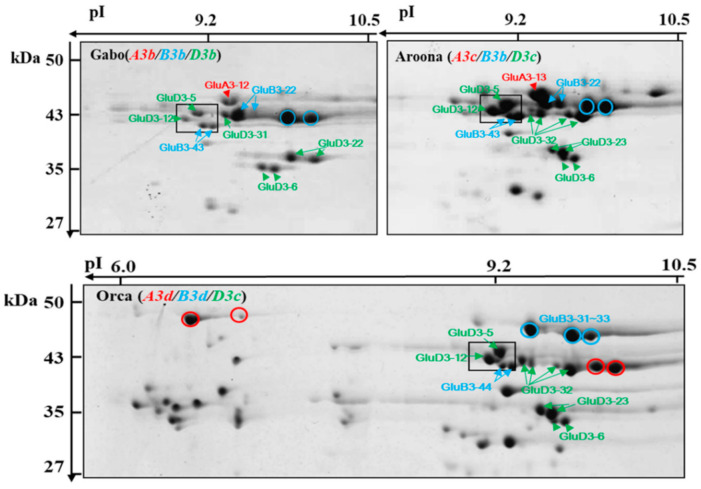
Two-dimensional gel electrophoresis (2-DGE) separation of LMW-GSs from three wheat cultivars: ‘Chinese spring’ (*Glu-A3a*/*B3a*/*C3a*), ‘Gabo’ (*Glu-A3b*/*B3b*/*C3b*), ‘Aroona’ (*Glu-A3c*/*B3b*/*C3c*), and ‘Orca’ (*Glu-A3d*/*B3d*/*C3c*). Protein spots corresponding to each LMW-GS allele encoded by *Glu-A3*, *Glu-B3*, and *Glu-C3* are indicated by red, light blue, and green colors, respectively, based on integrated comparative analysis and identification using MS/MS.

**Table 1 ijms-22-07709-t001:** Allelic compositions of LMW-GS at the *Glu-A3*, *Glu-B3*, and *Glu-D3* loci in ‘Aroona’ and ‘Aroona’ near isogenic lines (ARILs).

Line	*Glu-A3*	*Glu-B3*	*Glu-D3*	Donor Parent
Aroona	***c***	*b*	*c*	Aroona
ARIL 15-4	***a***	*b*	*c*	Chinese spring
ARIL 16-1	***b***	*b*	*c*	Gabo
ARIL 18-5	***d***	*b*	*c*	Orca
ARIL 19-2	***e***	*b*	*c*	Lerma Rojo
ARIL 20-1	***f***	*b*	*c*	Bungulla
ARIL 21-2	*c*	***a***	*c*	Chinese spring
ARIL 23-4	*c*	***c***	*c*	Halberd
ARIL 24-3	*c*	***d***	*c*	Orca
ARIL 27-6	*c*	***g***	*c*	Millewa
ARIL 28-4	*c*	***h***	*c*	Sonalika
ARIL 36-2	*c*	*b*	***b***	Bungulla
ARIL 35-1	*c*	*b*	***f***	India 115

**Table 2 ijms-22-07709-t002:** Identification of LMW-GSs at the Glu-A3, Glu-B3, and Glu-D3 loci in ‘Aroona’ and ‘Aroona’ near isogenic lines (ARILs).

# Spot	Cultivars	Gene	MS/MS Identification	^a^ Gene Haplotype	N-Terminal Sequence	# A.A	^b^ Putative Corresponding Genes	
							Accession No. (Identity)	Gene
1	AROONA	*Glu-B3b*	ACA63873	*GluB3-22*	MENSHIP	349	EU369721 (100%)	*B3-621a*
2	AROONA	*Glu-B3b*	ACA63873	*GluB3-22*	MENSHIP	349	EU369721 (100%)	*B3-621a*
3	AROONA	*Glu-D3c*	ABC84367	*GluD3-32*	IENSHIP	334	FJ755316 (100%)	*D3-578a*
4	AROONA	*Glu-D3c*	ABC84367	*GluD3-32*	IENSHIP	334	FJ755316 (100%)	*D3-578a*
5	AROONA	*Glu-D3c*	ABC84367	*GluD3-32*	IENSHIP	334	FJ755316 (100%)	*D3-578a*
6	AROONA	*Glu-D3c*	ABC84367	*GluD3-32*	IENSHIP	334	FJ755316 (100%)	*D3-578a*
7	ARIL 27-6	*Glu-B3g*	ACA63857	*GluB3-23*	MENSHIP	349	EU369705 (100%)	*B3-621b*
8	ARIL 27-6	*Glu-B3g*	ACA63857	*GluB3-23*	MENSHIP	349	EU369705 (100%)	*B3-621b*
9	ARIL 27-6	*Glu-B3g*	ABY58126	*GluB3-15*	MENSHIP	323	EU369703 (100%)	*B3-544*
10	ARIL 27-6	*Glu-B3g*	ABY58126	*GluB3-15*	MENSHIP	323	EU369703 (100%)	*B3-544*
11	ARIL 27-6	*Glu-B3g*	ACZ59817	*-*	MENSHIP	324	EU369703 (100%)	*B3-544*
12	ARIL 36-2	*Glu-D3b*	ABC84366	*GluD3-31*	MENSHIP	344	JX878006 (100%)	*D3-578b*

^a^ LMW-GS genes identified from *Glu-B3* [34] and *Glu-D3* [35] ^b^ LMW-GS genes isolated from ‘Aroona’ and ARILs [19,36].

**Table 3 ijms-22-07709-t003:** Active LMW-GS genes at the *Glu-A3*, *Glu-B3*, and *Glu-D3* in ‘Aroona’ and ‘Aroona’ near-isogenic lines (ARILs).

Line	Alleles	^a^ Active LMW-GS Genes (^b^ Gene Name in Chinese Wheat Germplasm)					
At *Glu-3A*							
ARIL15-4	*Glu-A3a*			*GluA3-11* *(A3-620)*			
ARIL16-1	*Glu-A3b*			*GluA3-12* *(A3-643)*			
Aroona	*Glu-A3c*			*GluA3-13* *(A3-620)*			
ARIL18-5	*Glu-A3d*	*GluA3-23* *(A3-402)*	*-* *(A3-568)*	*GluA3-4* *(A3-662)*			
ARIL19-2	*Glu-A3e*			*GluA3-15*/*17* *(A3-646)*			
ARIL20-1	*Glu-A3f*		*GluA3-16* *(A-573)*				
At *Glu-3B*							
ARIL21-2	*Glu-B3a*	*GluB3-44* *(B3-530a)*	*GluB3-11* *(B3-593)*	*GluB3-21* *(B3-624)*			
Aroona	*Glu-B3b*	*GluB3-43* *(B3-530b)*	*GluB3-12* *(B3-607)*	*GluB3-22* *(B3-621a)*			
ARIL23-4	*Glu-B3c*	*GluB3-44* *(B3-530a)*		*GluB3-34* *(B3-688a)*			
ARIL24-3	*Glu-B3d*	*GluB3-44* *(B3-530a)*		*GluB3-31–33* *(B3-688b)*			
ARIL27-6	*Glu-B3g*	*GluB3-41* *(B3-530c)*	*GluB3-15* *(B3-544)*	*GluB3-23* *(B3-621b)*			
ARIL28-4	*Glu-B3h*	*GluB3-43* *(B3-530b)*		*GluB3-61–64* *(B3-688c)*			
At *Glu-3D*							
ARIL36-2	*Glu-D3b*	*GluD3-6* *(D3-385)*	*GluD3-4* *(D3-394)*	*GluD3-22* *(D3-441)*	*GluD3-12* *(D3-528)*	*GluD3-5* *(D3-575)*	*GluD3-31* *(D3-578b)*
Aroona	*Glu-D3c*	*GluD3-6* *(D3-385)*	*GluD3-4* *(D3-394)*	*GluD3-23* *(D3-432)*	*GluD3-12* *(D3-528)*	*GluD3-5* *(D3-575)*	*GluD3-32* *(D3-578a)*
ARIL35-1	*Glu-D3f*	*GluD3-6* *(D3-385)*	*GluD3-4* *(D3-394)*	*GluD3-22* *(D3-441)*	*GluD3-12* *(D3-528)*	*GluD3-5* *(D3-575)*	*GluD3-31* *(D3-578b)*

^a^ LMW-GS genes identified from *Glu-A3* [37], *Glu-B3* [34], and *Glu-D3* [35] ^b^ LMW-GS genes isolated from ‘Aroona’, ARILs, and Chinese wheat germplasm [19,36].

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
