# Peer review of "Proteomic Determination of Low-Molecular-Weight Glutenin Subunit Composition in Aroona Near-Isogenic Lines and Standard Wheat Cultivars"

_ijms, 2021, doi:10.3390/ijms22147709_

Round 1

Reviewer 1 Report

Article "Proteomic Determination of Low-Molecular-Weight Glutenin 2
Subunit Composition in Aroona Near-Isogenic Lines and 3
Standard Wheat Cultivars" interesting and will attract wheat breeders. LMW-GS alleles in germplasm remain an important challenge for wheat breeding and marker-assisted selection based on favorable alleles is very important for quality breeding of wheat. The paper is well written in a convincing style and I believe that such type of paper will be good for readers and the journal as well.

Author Response

Thank you for reviewing our manuscript.

Reviewer 2 Report

This study reports the identification of different LMW-GS in one wheat cultivar Aroona and 12 associated near-isogenic lines. This is a very interesting paper, especially since the LMW-GS always seem to be less studied compared to HMW-GS.

Comments

L42: Remove one hyphen in “inter-molecular”

L65: Usually, this method is abbreviated as UHPLC, unless the instrument is from the manufacturer Waters (they had trademarked UPLC)

L70: 2DGE has already been introduced

L77: NIL / ARIL: Both abbreviations are used for near-isogenic lines, but it is not clear, why both are needed. Please harmonize.

L83: Is this statement still true, with the wheat genome reference sequence available now?

Figure 2: Why do some gels show the pH range from 6.0 to 10.5 and others only the small range from 8? To 10.5? It appears that the neutral/slightly acidic pH range does not show LMW-GS.

Figure 3: Why does the legend state four wheat cultivars? It appears that only three are shown here.

Reference list, ref. 1: First author is called Shewry

Author Response

Thank you for your comments to improve our manuscript. Based on your comments, We revised them as follows.

L42: Remove one hyphen in “inter-molecular”

Changed “inter--molecular” to “inter-molecular”

L65: Usually, this method is abbreviated as UHPLC, unless the instrument is from the manufacturer Waters (they had trademarked UPLC)

Changed “RP-HPLC” to “RP-UHPLC” in L65 and L263

L70: 2DGE has already been introduced

Changed “two-dimensional gel electrophoresis (2-DGE)” to “2-DGE” in L70

L77: NIL / ARIL: Both abbreviations are used for near-isogenic lines, but it is not clear, why both are needed. Please harmonize.

Changed “near-isogenic lines (NILs)” to “near-isogenic lines” in L77

Changed “NILs” to “near-isogenic lines” in L162

Changed “near-isogenic lines (NILs)” to “near-isogenic lines” in L426

L83: Is this statement still true, with the wheat genome reference sequence available now?

Thank you for mentioning what I missed. The sentence is changed to " This is a result of a lack of efficient techniques for isolating individual LMW-GS genes from the multiple highly similar LMW-GS genes within a cultivar, and the allelic variation between cultivars."

Figure 2: Why do some gels show the pH range from 6.0 to 10.5 and others only the small range from 8? To 10.5? It appears that the neutral/slightly acidic pH range does not show LMW-GS.

Actually, LMW-GSs in Aroona and 12 ARILs were separated on 2-DGE with the same condition (pH6~11) as mentioned in Materials and Methods. Most LMW-GS spots were observed between pI 8.0 and 10.5 except for ARIL 18-5(Glu-A3d) where acid LMW-GSs were observed. So, in order to compare patterns of LMW-GS in ARIL 18-5 with those in Aroona, the broad pH range of 2-DGE in two cultivars was shown. However, because LMW-spots in the other ARILs gathered on a short pH range, we used only the certain pH range (pH8.0~10.5) to present easily and clearly the comparative analysis among ARILs.

Figure 3: Why does the legend state four wheat cultivars? It appears that only three are shown here.

Changed “four” to “three” in the legend of Figure 3

Reference list, ref. 1: The first author is called Shewry

Changed “Shwry” to “Shewry” in Ref. 1